



# Ideas and perspectives: Land-ocean connectivity through groundwater

Damian L. Arévalo-Martínez [1,2,*] Amir Haroon [1,*], Hermann W. Bange [1], Ercan Erkul [2], Marion Jegen [1], Nils Moosdorf [2,3], Jens Schneider von Deimling [2], Christian Berndt [1], Michael Ernst Böttcher [4,5,6], Jasper Hoffmann [7], Volker Liebetrau [1,†], Ulf Mallast [8], Gudrun Massmann [9], Aaron Micallef [1,10], Holly A. Michael [11], Hendrik Paasche [8], Wolfgang Rabbel [2], Isaac Santos [12], Jan Scholten [2], Katrin Schwalenberg [13], Beata Szymczycha [14], Ariel T. Thomas [10], Joonas J. Virtasalo [15], Hannelore Waska [9], Bradley Weymer [16].

[1] GEOMAR Helmholtz Centre for Ocean Research Kiel, Kiel, 24105, Germany
[2] Kiel University, Kiel, 24118, Germany
[3] Leibniz Centre for Tropical Marine Research (ZMT), Bremen, 28359, Germany
[4] Leibniz Institute for Baltic Sea Research Warnemünde (IOW), Rostock, 18119, Germany
[5] Marine Geochemistry, University of Greifswald, Greifswald, 17489, Germany
[6] Interdisciplinary Faculty, University of Rostock, Rostock, 18051, Germany
[7] Alfred-Wegener-Institute, Helmholtz Centre for Polar and Marine Research, Bremerhaven, 27515, Germany
[8] Helmholtz Centre for Environmental Research, Leipzig, 04318, Germany
[9] Carl von Ossietzky University of Oldenburg, Oldenburg, 26129, Germany
[10] University of Malta, Msida, MSD 2080, Malta
[11] University of Delaware, Newark, DE 19716, USA
[12] Department of Marine Science, University of Gothenburg, Gothenburg, 40539, Sweden
[13] Federal Institute for Geosciences and Natural Resources, Hannover, 30655, Germany
[14] Institute of Gdańsk Polish Academy of Sciences, Sopot, 81-712, Poland
[15] Marine Geology, Geological Survey of Finland (GTK), Espoo, 02150, Finland
[16] School of Oceanography, Shanghai Jiao Tong University, China
[†] deceased

[*] *Correspondence to*:

Damian L. Arévalo-Martínez (darevalo@geomar.de) or Amir Haroon (aharoon@geomar.de)

**Abstract.** For millennia humans have gravitated towards coastlines for their resource potential and as geopolitical centres for global trade. A basic requirement ensuring water security for coastal communities relies on a delicate balance between the supply and demand of potable water. The interaction between freshwater and saltwater in coastal settings is, therefore, complicated by both natural and human-driven environmental changes at the land-sea interface. In particular, ongoing sea level rise, warming and deoxygenation might exacerbate such perturbations. In this context, an improved understanding of the nature and variability of groundwater fluxes across the land-sea continuum is timely, yet remains out of





reach. The flow of terrestrial groundwater across the coastal transition zone as well as the extent of freshened groundwater below the present-day seafloor are receiving increased attention in marine and coastal sciences because they likely represent a significant, yet highly uncertain component of (bio)geochemical budgets, and because of the emerging interest in the potential use of offshore freshened groundwater as a resource. At the same time, "reverse" groundwater flux from offshore to onshore is of prevalent socio-economic interest as terrestrial groundwater resources are continuously pressured by overpumping and seawater intrusion in many coastal regions worldwide. An accurate assessment of the land-ocean connectivity through groundwater and its potential responses to future anthropogenic activities and climate change will require a multidisciplinary approach combining the expertise of geophysicists, hydrogeologists, (bio)geochemists and modellers. Such joint activities will lay the scientific basis for better understanding the role of groundwater in societal-relevant issues such as climate change, pollution and the environmental status of the coastal oceans within the framework of the United Nations Sustainable Development Goals. Here, we present our perspectives on future research directions to better understand land-ocean connectivity through groundwater, including the spatial distributions of the essential hydrogeological parameters, highlighting technical and scientific developments, and briefly discussing its societal relevance in rapidly changing coastal oceans.

## 1 Background

The exchange of groundwater between land and ocean is a wide-spread phenomenon, which has significant impacts on the biogeochemical cycles of the coastal ocean (e.g. Church, 1996; Moore, 2010; Santos et al., 2021). Coastal margins play a disproportionally important role for productive marine ecosystems compared to the open ocean due to their greater biological productivity, sediment-water interactions and air-sea transfer of climate-relevant trace gases (Liu et al., 2010). Increasing anthropogenic activities result in high nutrient fluxes into the coastal ocean, leading to eutrophication, deoxygenation and release of greenhouse gases, which in turn could exacerbate the current global warming trend and significantly affect the livelihood of nations that rely on coastal ecosystem services (e.g. Van Meter et al., 2018; Oehler et al., 2021; Rocha et al., 2021). In addition, accelerating global sea level rise (GSLR) can negatively influence terrestrial coastal aquifers due to the inland displacement of the fresh-saline-water interfaces, referred to as saltwater intrusion (SWI; Ferguson and Gleeson, 2012; Taylor et al., 2013). In turn, increased human usage of groundwater resources is estimated to account for approximately 14% of the observed GSLR through a net transfer of freshwater from deep reservoirs into the ocean (Konikow, 2011; Church et al., 2013; Taylor et al., 2013). Increasing usage of non-renewable groundwater might further exacerbate global water depletion (Bierkens and Wada, 2019), which is further impacted by climate variability through changes in recharge and precipitation (Thomas and Famiglietti, 2019; Beebe et al., 2022).

The cross-shelf extension of terrestrial coastal groundwater systems can be distinguished into two key (often interrelated) elements (see Table 1 and Fig. 1). The first comprises meteoric groundwater transport (flux) from terrestrial coastal aquifers through the seafloor into the ocean, which is generally referred to as fresh submarine groundwater discharge (FSGD; e.g. Kohout, 1964; Taniguchi et al., 2019). The second





consists of large (> 10 km horizontal extent) freshened (and often brackish) groundwater reservoirs embedded in sediment and rocks below the present-day seafloor, collectively called offshore freshened groundwater (OFG; Post et al., 2013).


FSGD connects terrestrial groundwater systems to the coastal ocean on most coastlines in the world (Fig. 2; Luijendijk et al., 2020). FSGD is essentially the surplus of the terrestrial water budget. Most known FSGD occurs within the first few 100 meters from the coast, although its occurrence has also been reported at tens to hundreds of kilometres offshore (Manheim, 1967; Kooi et al., 2001; Bratton et al.,

2010). Given the large degree of spatio-temporal variability in FSGD, estimates of regional and global fluxes are still highly uncertain (Taniguchi et al., 2019). Globally, FSGD accounts for 1–10 % of the global freshwater input to the ocean (Abbott et al., 2019; Luijendijk et al., 2020). Locally, however, FSGD can be key for sustaining some marine ecosystems (Luijendijk et al., 2020).

Similar to FSGD, OFG has been observed in shelf sediments throughout the world's oceans (Fig.2; Post et al., 2013; Micallef et al., 2021). Likewise, OFG is a potential freshwater resource, or a resource of water that can be treated with desalination with comparably small energy consumption (Bakken et al., 2012), and has therefore gained increased attention over the past decade (Post et al., 2013; Micallef et al., 2021). Although OFG is generally a relic of past sea-level low stand (fossil groundwater), some reservoirs

are likely hydraulically connected to the terrestrial aquifers groundwater system, as shown for the U.S. Atlantic coast (Gustafson et al., 2019; Thomas et al., 2019), Canterbury Bight, New Zealand (Micallef et al., 2020; Weymer et al., 2020), and the Achziv submarine canyon in northern Israel (Paldor et al., 2020). Here, we emphasize the importance of improving our understanding of connected OFG, since its extraction as an unconventional resource for mitigating temporal water scarcity in coastal communities

might cause seawater intrusion (Yu and Michael, 2019a) and distant land subsidence (Chen et al., 2007; Yu and Michael, 2019b).

With ongoing research in near-coastal groundwater fluxes (FSGD) and offshore reservoirs (OFG) carried out by largely different scientific communities, we address unexploited scientific and technical synergies

between them. The reliance on markedly different methodologies leads to differences in scientific language, and in turn conceptually disconnects the research of both phenomena (FSGD is usually assessed using geochemical tracers and hydrological observations from the intertidal zone or numerical groundwater modelling (Taniguchi et al., 2019; Luijendijk et al., 2020), whereas OFG studies often require ship-based geophysical methods (Micallef et al., 2021)). Here we present new perspectives on

future research directions to improve the understanding of land-ocean connectivity through groundwater, with particular focus on joint activities of FSGD and OFG research communities. This includes *i)* improving our quantitative understanding of the distribution and variability of groundwater fluxes at regional and global scales, *ii)* assessing long-term changes in groundwater sources and their expected impact on marine environments, as well as potential usage, and *iii)* evaluating conceptual and

technological developments which will potentially advance joint FSGD-OFG research.



## 2 Distribution and variability of groundwater fluxes

*Fresh submarine groundwater discharge*

The interaction between saline and fresh groundwater in coastal regions is governed by complex processes, e.g. density contrasts between fresh and saline water, tidal effects, and geological heterogeneity (Michael et al., 2016; Jiao and Post, 2019). Saline groundwater can intrude landward salinizing terrestrial aquifers (resulting in SWI). Yet, at the same time, terrestrial groundwater can cross the land-sea continuum and appear offshore as FSGD and/or OFG (Fig. 2; see e.g. Whiticar, 2002; Post et al. 2013; Jurasinski et al., 2018; Micallef et al. 2020). Groundwater flow is associated with external forcing (e.g. groundwater heads, framework geology, onshore groundwater usage, sea level) that dictate the hydrostatic gradient causing fluxes to be directed inland, offshore or both. Strong distortions of hydraulic gradients can influence or even reverse groundwater flow, which, in turn, might have widespread consequences for pelagic and benthic marine ecosystems (e.g. Donis et al., 2017; Lecher and Mackey, 2018; Szymczycha et al., 2020; Santos et al., 2021), as well as for associated services such as fisheries, because both nutrients and contaminants are transported into the coastal ocean via groundwater. A recent study estimated the global input of groundwater into the ocean via FSGD to be less than 1% of the surface-water runoff. However, on local scales FSGD can reach 25% of the river flux (Luijendijk et al., 2020), and saline SGD releases recycled nutrients at rates comparable to global rivers (Santos et al., 2021). The high spatial variability of this influx is partly controlled by climate at regional scale, and partly by lithological heterogeneities at local scale (Sawyer et al., 2016). Because the extrapolation of point-scale measurements onto a regional, continental, or global scale is difficult, FSGD quantification heavily relies on hydrogeological modelling (Moosdorf et al., 2021), which can result in great uncertainties on large spatial scales.

*Offshore freshened groundwater*

OFG resides beneath the seafloor along continental shelves and, in contrast to FSGD, is commonly assumed to have minimal groundwater flow velocities (e.g. Micallef et al. 2020). Recent estimates report OFG to comprise a volume of approximately $1*10^6$ km$^3$ (Micallef et al., 2021). Different OFG emplacement mechanisms have been proposed, from which meteoric recharge, sub- and proglacial injection, diagenesis and the decomposition of gas hydrates are the most significant (Micallef et al., 2021). OFG systems may be coupled with FSGD (e.g. Paldor et al., 2020; Attias et al., 2021), and modelling shows that FSGD and OFG can occur in equilibrium with present-day sea level for a range of different stratigraphic configurations (Michael et al., 2016). However, OFG can also be decoupled from interaction with the water column (see e.g. Micallef et al., 2020). Post el al. (2013) compiled a global estimation of OFG sites based mainly on borehole observations. Geophysical technologies have updated these global estimates through the detection of OFG residing within siliciclastic continental margins in the United States and New Zealand (Gustafson et al., 2019; Micallef et al., 2020), along a carbonate coastline in Malta (Haroon et al., 2021), and offshore from the volcanic islands of Hawaii (Attias et al., 2021). These studies have improved our understanding of spatial OFG distributions, but do not bridge the knowledge gap between coastal nearshore and offshore hydrological systems. To date, continuous tracing of terrestrial aquifers along the full onshore-offshore gradient remains technically challenging (Weymer et





al., 2020), and observation strategies need to be developed for specific settings. Geophysical methods employed as imaging tools to characterize the subsurface offer promising avenues towards bridging the information gap across the land-sea interface, although they are only currently available on local scales (e.g. Siemon et al., 2020; Ishizu and Ogawa, 2021). Hydrologically connected OFG systems should in

principle be associated with discharging groundwater (see e.g. Weymer et al., 2020), either close to the coastline, along faults or other lithological discontinuities, or at distant locations near the shelf break. However, OFG could also seep into the marine environment on time scales of >100−1000 kyr, making it difficult to obtain observations that provide insights on its effects on biological communities if no dedicated offshore drilling is carried out.


## 3 Environmental impacts and resource prospects

In the terrestrial realm, the role of coastal groundwater as a habitat (Pohlman, 2011; Leitão et al., 2015; Adyasari et al., 2019) and in shaping pelagic and benthic coastal communities (Lecher and Mackey, 2018;

Oberle et al., 2022) has become increasingly recognized. In contrast, the role of OFG as a fresh- or brackish water habitat within a purely marine environment remains unknown, and might constitute a new frontier in ocean sciences.

Human use of OFG could affect both fresh groundwater discharging into the ocean and groundwater

hydraulic heads on land. FSGD has local ecological impacts on e.g. seagrass (Carruthers et al., 2005), corals (Oehler et al., 2019; Correa et al., 2021; Oberle et al., 2022), phytoplankton (Rodellas et al., 2015; Sugimoto et al., 2017; Waska and Kim, 2010), mollusc (Hwang et al., 2010), meio/macrofauna (Zipperle and Reise, 2005; Kotwicki et al., 2014; Grzelak et al., 2018) and fish populations (Fujita et al., 2019; Pisternick et al., 2020). These influences are often triggered by nutrient and carbon inputs into the

submarine environment (Santos et al., 2021; Böttcher et al., 2022). Moreover, upward fluid migration within soft seafloor sediments might fluidize them, favouring the formation of pockmarks that potentially release greenhouse gases such as carbon dioxide and methane (Whiticar, 2002; Judd and Hovland, 2009; Donis et al., 2017; Virtasalo et al., 2019; Hoffmann et al., 2020). While some of these effects have been perceived as a threat for ecosystems, for instance by inducing toxic algal blooms, adding alkalinity (Cabral

et al., 2021) or harbouring dense microbial communities (Ionescu et al., 2012), they can also sustain coastal ecology and increase fishery yields. Pumping OFG that is associated with FSGD could reduce the associated landward reservoirs and eventually impact the coastal marine environment. Moreover, anthropogenic intervention on coastal sediments might impact benthic-pelagic coupling associated with FSGD (von Ahn et al., 2021).


Considering the manifold biogeochemical impacts of FSGD, it is difficult to assess the overall effect of different pumping approaches and particular FSGD locations in local marine ecosystems, should a connected OFG be exploited. Pumping water from a groundwater system means reducing the formation pressure. The reduced pressure can communicate to the terrestrial aquifer and reduce the hydraulic head

there (Yu and Michael, 2019b). The extent of this effect will depend on the reservoir properties as well as the hydraulic connectivity between terrestrial and offshore domain, which might in turn lead to SWI (Ferguson and Gleeson, 2012; Yu and Michael, 2019a), groundwater depletion (Bierkens and Wada,



2019) and subsidence (Yu and Michael, 2019b). Despite the large uncertainties on the global and long-term effects of these changes for groundwater resources and associated marine ecosystems, lessons may
be learned from the environmental effects of extensive oil exploration (Varma and Michael 2012; Chaussard et al., 2013).

Changes in FSGD volume and its chemical/biological composition could serve as an important indicator for changes in the coastal groundwater system, which could, in turn, also be caused by connectivity with
OFG. FSGD can be a source of geochemical tracers (e.g. Ra and Rn; Kim and Hwang, 2002), inorganic nutrients (nitrate, phosphate and silicate; e.g. Waska et al., 2011; Szymczycha et al., 2012), trace metals (e.g. Knee and Paytan, 2011), climate-relevant trace gases (carbon dioxide, nitrous oxide, methane and carbon monoxide; e.g. Bugna et al., 1996; Chapelle and Bradley, 2007; Jurado et al., 2017; Kolker et al., 2021; Reading et al., 2021) and organic material (e.g. dissolved organic matter; see Kim and Kim (2017)
and McDonough et al. (2022)) to coastal areas. The input of nutrients results in a FSGD-driven eutrophication of coastal areas and, thus, potentially affects coastal ecosystems (Luijendijk et al., 2020; Oehler et al., 2021; Santos et al., 2021). For example, the large *Ulva prolifera* outbreaks (green tides), which occur regularly off the coast of China, are attributed to the nutrient supply by FSGD (Liu et al., 2017; Zhao et al., 2021). Hence, sustained monitoring the biogeochemical and microbially-driven
transformations of key biogeochemical tracers within the subterranean estuary as well as their release to the overlying water column, might help tracking changes in FSGD. Furthermore, such monitoring might also facilitate investigating potential impacts on the productivity and ecological status of coastal environments. Beyond coastal nearshore environments, FSGD seems to play an important role for biogeochemical fluxes to the ocean and affects benthic and sub-seafloor ecosystems in more offshore
coastal areas (Micallef et al., 2021 and references therein). Therefore, a thorough investigation of its dynamics in different oceanic basins and geological settings should be performed by future studies.

Furthermore, FSGD and connected OFG could be increasingly affected by ongoing environmental changes on the terrestrial side namely climate change (e.g. by changing rain patterns or intensity; Thomas
and Famiglietti, 2019), eutrophication (derived from increasing applications of fertilizers), urbanization of coastal areas and associated contamination with microplastics (Viaroli et al., 2022), chemical (e.g. pesticides, pharmaceuticals, and personal care products; see Knee and Paytan, 2011; Szymczycha et al., 2020) and biological pollutants (pathogenic germs such as bacteria and viruses; see e.g. Kyle et al., 2008; Sorensen et al., 2021). In particular, the effects of FSGD-driven inputs of chemical/biological pollutants
on coastal areas remain largely unknown.

## 4 Conceptual and technological approaches for assessing land-ocean groundwater connectivity

Various techniques are available to explore and identify FSGD and OFG in the offshore environment (e.g.
Micallef et al., 2021) and groundwater resources on the land side (Kirsch, 2006). These techniques often reveal anomalies in the subsurface, the seafloor (e.g. pockmarks) or sea water column (e.g. salinity, geochemical tracers) associated with fresh groundwater. Often multiple techniques are applied to build confidence in interpretation of groundwater dynamics. The technologies used can be broadly categorized in four groups: *i)* geophysical imaging techniques, which detect/record physical parameters such as



electrical resistivity, seismic velocity, density, temperature or structural/morphological surface anomalies, *ii)* hydrogeological approaches, including modelling and hydrological measurements (e.g. hydraulic heads, salinity, and recharge rates) *iii)* (bio)geochemical techniques, which analyse (bio)geochemical fingerprints of the fluids, and *iv)* (micro)biological sampling, which unravels biological diversity and processes associated with FSGD. These multiple approaches are often mastered by
researchers within different disciplinary backgrounds including geophysics, hydrology, oceanography and biogeochemistry.

Assessing land-ocean hydraulic connectivity through groundwater requires investigating the connectivity of underlying lithologic units and their hydrological characterization. Moreover, it also requires
identifying the current distribution of freshened groundwater bodies across the coast line. The occurrence of freshened groundwater along the onshore-offshore continuum may in turn be read from geochemical fingerprinting of fluid samples obtained from the different realms. Hence, the success of such a highly interdisciplinary endeavour in mapping and understanding the connectivity will depend on how well the different methodologies can be integrated. Here, we suggest overarching approaches in which synergies
(both conceptual and technological) between FSGD and OFG scientific communities could contribute to an improved understanding of the dynamics of groundwater as a connecting path between land and the ocean at the coastal zone. Table 2 presents some of the most commonly used methods in groundwater studies, for which we foresee promising synergies between FSGD and OFG research.

*Shoreline-crossing lithologies*
Seismic reflection imaging is the method of choice for detailed subsurface mapping. Particular lithologies may be identified by the character of the seismic reflection data within a lithological unit, e.g. layered seismic facies for fine-grained marine sediments vs. chaotic patterns for coarser grained sediments (Thomas et al., 2019; Micallef et al., 2020). Co-located boreholes on seismic sections can greatly improve
the identification of different facies and serve as calibration points along those sections. Of particular importance for shoreline-crossing groundwater dynamics is the possibility of seismic data to constrain the continuity of different lithological units, the presence of impermeable clay layers and faults, or other disrupting geological structures. However, seismic information in the transition zone near the coastline is not widely available due to logistical challenges for data acquisition. Land and marine seismic data have
inherently different signal-to-noise ratios and imaging depths, making across-shoreline interpretation challenging. On land there are often more boreholes than offshore, which can provide data to constrain the lithology distribution. Through the integration of onshore and offshore seismic and borehole data using geostatistical methods such as sequential indicator simulation or multiple point geostatistics (Deutsch and Pyrcz 2014), lithology distribution across the shoreline can be modelled to reduce the
uncertainty of connected pathways between the terrestrial and offshore domains.

Land and marine seismic data require different seismic sources, e.g. vibroseis on land and airguns/sparkers at sea, and receivers. Noise levels are generally higher on land, whereas offshore imaging in the transition zone is hampered by seafloor multiple reflections due to the shallow water depth.
Generally, clastic sedimentary environments are easier to image than carbonate systems (e.g. Mountain, 2008; Lofi et al., 2013; Bertoni et al., 2020). Amphibious data acquisition, i.e. across the shoreline, is





possible and can be accomplished in different ways, for instance by shooting on land and receiving at sea or vice versa. Yet, due to logistical challenges and greater expenses, amphibious sections are not a standard. In karstic carbonate or volcanic systems, the spatial occurrence of localised submarine springs
(rather than the diffuse discharge in siliciclastic systems) can help to characterise the onshore-offshore connectivity of aquifers (Bayari et al., 2011).

Ground penetrating radar (GPR) is suitable for detailed, near-surface lithological imaging on land. The GPR technique is based on an electromagnetic signal that is sensitive to sediment water content. Offshore,
hydroacoustic and seismic methods provide structural information from shallow to larger depth, but are insensitive to the water content. However, unconformable boundaries of subsurface sediment units are typically imaged as strong reflectors in both GPR and reflection seismic profiles due to the associated sharp changes in water content and density, respectively, which permits the cross-shore correlation of onshore GPR profiles with marine seismic profiles using the allostratigraphic approach (see e.g. Virtasalo
et al., 2019; Peterson et al., 2020).

*Identification of ground water bodies*
While seismic data can reveal the geological background and are -to some extent- sensitive to the porosity of the rock, they contain no information on pore fluid salinity. The salinity of pore fluids can be explored
using electrical methods because the bulk electrical resistivity of a sediment rock is governed by the amount (fluid-saturated pore space) and salinity of fluid present (Archie, 1942; Keller, 1987). The better the porosity of the lithology is known, for example through seismic/lithological data, the better the pore space fluid saturation on land and the pore water salinity offshore can be assessed from bulk electrical resistivity measurements. A bulk electrical resistivity model of the subsurface can be derived from either
direct or alternating current electrical measurements (electromagnetic induction), where the latter allows for larger penetration depths and better resolution offshore.

On land, the highest data acquisition speed and therefore the largest areal coverage is achieved through airborne electromagnetic methods (e.g. Bedrosian et al., 2016; Gottschalk et al., 2020; Siemon et al.,
2020). Additional ground measurements using direct current and controlled source electromagnetic methods provide bulk electrical resistivity model of the subsurface at higher resolution and larger depths of penetration (e.g. Pondthai et al., 2020). Resolution in surveys of electrical resistivity on land can be augmented by conducting GPR surveys. GPR methods allow both detecting contrasts in the electrical conductivity structure (dielectric constant) contained in coastal sediments at high resolution (cm to m
scales), and effectively mapping the freshwater-saltwater interface at shallow depths (up to tens of m; Weymer et al., 2020).

Offshore, a freshened groundwater body offshore can be identified as an electrical resistivity anomaly caused by the resistivity contrast between fresh and saline pore water. The conductive saline ocean above
the seafloor strongly damps electromagnetic signals which renders airborne electromagnetic systems incapable of penetrating the seafloor at water depths larger than about 10–20 m (Goebel et al., 2019). Therefore, offshore measurements require specially adapted marine electromagnetic systems. So far, OFG exploration studies have been conducted using surface-towed (e.g. Gustafson et al., 2019; Attias et al.,





2021) and/or seafloor-towed (Haroon et al., 2018; 2021; Micallef et al., 2020) systems. Both systems
consist of a horizontal electric source dipole followed by several electric receiving dipoles recording the inline electric field. Offsets between transmitter and receiving dipoles typically range can between hundreds and several hundreds of meters, and can be adjusted according to the target depth. Sea surface-towed systems have the advantage of a greater acquisition speed, yet at the cost of lower resolution and larger source dipole moments (current amplitude times dipole length) required to compensate for the
decay of the source signal in the conductive ocean layer. Seafloor-towed systems have arguably better signal to noise ratios and resolution, although survey speed is much lower, and surveying is hampered by rough seafloor topography and infrastructure. Onshore-offshore acquisition with a land transmitter and offshore receiver is possible (Ishizu and Ogawa, 2021). However, to date there are no peer-reviewed published studies which use this approach. Merging of a separately acquired onshore-offshore electrical
resistivity section with land and marine systems is possible, although a coherent continuous picture may be hampered by different resolutions, penetration depths, noise levels and the strong 3D resistivity contrast at the shoreline ("coast effect"; Worzewski at al. (2012)). Recently, joint land/water data inversion methods have become available to amend that deficit (Hermans and Paepen, 2020). Furthermore, conversion of electrical resistivity sections to water saturation on land or pore water salinity
(the actual target parameters), requires integration of lithological data, i.e. bulk porosity estimates and an appropriate choice of effective medium model.

In-situ sampling techniques are effective and simple, albeit labour- and time-intensive ways to detect freshened groundwater. These methods include pore water extraction using push-point samplers along
transects or grids (Waska et al., 2019), in-situ detection of springs with infrared cameras (Röper et al., 2014), and collection of seeping groundwater with seepage meters or benthic chambers (e.g. Lee 1977, Donis et al., 2017). Although mostly applied to nearshore groundwater discharge, all above-mentioned methods are adaptable to remote systems, for instance on stationary landers or ROVs (e.g. Ahmerkamp et al., 2017).

*Groundwater flow*
Imaging of coastal aquifers using inversion of geophysical data constrained by groundwater transport simulations is a promising method which might greatly reduce uncertainty in FSGD rates and location (Costall et al., 2020). Other promising approaches to detect FSGD over a larger area (tens of kilometres)
while also allowing an assessment of its temporal variability, are thermal radiance measurements with manned (e.g. Roxburgh, 1985; Johnson et al., 2008) or unmanned (e.g. Fischer et al., 1964; Dulai et al., 2016; Lee et al., 2016; Mallast and Siebert, 2019) aerial and sea-going vehicles.

While geophysical methods provide the geological background and current state of onshore-offshore
groundwater distribution (Weymer et al., 2015), they do not capture the dynamics and functioning of the system which are essential to determining and understanding the nature of land-sea hydrologic connectivity. Physical hydrological measurements are essential for understanding groundwater flow rates and patterns. In coastal systems with variations in fluid density, this involves characterizing groundwater head distributions and associated hydraulic gradients, as well as the salinity distributions. On land, this is
typically done with measurements from groundwater wells in addition to geophysics. Offshore, these





measurements are more challenging but provide critical information on the forces driving fluid flow through the onshore-offshore system. Because offshore hydrologic data is generally sparse, groundwater modelling is an essential tool to test hypotheses about system function given the geological, hydrological, and biogeochemical data available. Groundwater models that incorporate physics-based variable-density flow and salt transport, and capture the essential characteristics of the system (e.g. interconnection of geologic strata; Michael et al., 2016; Thomas et al., 2022), can be used to understand the long-timescale evolution of OFG systems towards their current state (e.g. Cohen et al., 2010; Micallef et al., 2020; Zamrsky et al., 2020), and to predict changes under expected changes in sea level and anthropogenic forcing (Yu and Michael, 2019a; 2019b). These models can not only characterize the flow in the subsurface, but also characterize the rate and distribution of FSGD and/or diffusive transport processes.

In conjunction with geophysical and hydrological data and analyses, the geochemistry of groundwater fluid samples can provide key information about the origin and age of FSGD and OFG. A combination of stable isotope and conservative tracer analysis (e.g. Hoefs, 2009; Dang et al., 2020) can be used to identify sources and estimate ages of offshore groundwater bodies, i.e. recent or fossil meteoric water (van Geldern et al., 2013), glacial meltwater (Hong et al., 2019) or methane hydrate dissociation (Dählmann and De Lange, 2003). While onshore fluid samples required for this analysis are relatively easily obtained (typically from groundwater observation wells), OFG fluid sample collection requires in-situ sampling at depth through a borehole or, if existing, knowledge of FSGD occurrences on the seafloor. FSGD sites on the seafloor can be identified through identification of morphologic depressions (pockmarks, sinkholes) through high frequency acoustic seafloor bathymetry mapping and identification of anomalous seafloor fauna and flora associated with a change in water salinity and nutrients input (e.g. Lecher and Mackey, 2018; Archana et al., 2021). Other approaches used to search FSGD sites on a regional scale include mapping radiogenic isotopes that are associated with groundwater (Burnett, 2006; Paldor et al., 2020), shallow physical imaging of resistivity anomalies, survey of small-scale magnetic susceptibility anomalies caused by preservation or diagenetic alteration of iron oxides in sediments (Müller et al., 2011), satellite infrared imagery using e.g. Landsat 8 - infrared (e.g. Wilson and Rocha, 2012; Schubert et al., 2014; Jou-Claus et al. 2021), and surface reaching fault mapping by seismic methods.

FSGD may also cause measurable anomalies in the deeper water column of offshore sites (Manheim, 1967; Attias et al., 2021). While temperature and salinity anomalies are only measurable in the immediate vicinity of the FSGD location and may be obscured by natural variations in water temperature and by tidal currents, Radon and Radium anomalies can be traced to larger distances (e.g. Cable et al., 1996; Moore et al., 2011). This methodology works well in areas of diffuse and uniform FSGD, but it might overlook localized point sources, which can account for up to 90% of FSGD in karstic regions (Null et al., 2014).

## 5 Future research directions

In view of the increasing pressure of human activities and natural changes on groundwater resources, the fundamental role of land-ocean connectivity through groundwater on the dynamics of coastal systems





requires a critical reassessment. FSGD and the associated fluxes of biogeochemical tracers might affect the physical structure, chemical composition and reactivity and the (micro)biology of the coastal ocean
ecosystems. Global and regional environmental changes (i.e. warming, eutrophication, acidification, pollution) modify processes in coastal groundwater and thereby FSGD, with largely unknown consequences for coastal marine ecosystems. Exploitation of OFG connected to terrestrial groundwater is expected to impact terrestrial groundwater flow systems. These feedback mechanisms operate over a wide range of spatial and temporal scales, ranging from molecular to global and from millisecond to
millennial. Thus, an overarching goal of future coastal groundwater research should aim to develop a suite of ecosystem models of land-ocean connectivity that include physical, geological, chemical and biological processes at play, and that address potential responses to dynamic interactions between nature and humans.

Within this framework, we recommend the following priority research tasks:

(1) assess and compare the spatio-temporal variability of physical and biogeochemical processes driving the dynamics of FSGD and OFG in different geological settings,

(2) characterize and quantify the geochemical/biological composition of FSGD and OFG, as well as its impacts on marine habitats and (micro)biological communities,

(3) develop an interdisciplinary framework including hydrological, geophysical, geochemical and (micro)biological and measurements to delineate groundwater fluxes (FSGD) and map reservoirs (OFG)
along the transition from nearshore to offshore systems,

(4) use numerical models and artificial intelligence to predict locations, magnitudes and connectivity of FSGD and OFG,

(5) characterize the stratigraphy at the land-ocean interface to determine the potential for development of connected, active OFG systems, and

(6) identify, quantify, and predict feedbacks between coastal groundwater dynamics and climate change to assess potential changes in volume and composition of FSGD and OFG.

Investigation of the land-ocean connectivity through groundwater beyond nearshore FSGD remains especially challenging because of its limited accessibility and large heterogeneity. Its future study will require representative and standardized sampling, the development of new analytical methods (e.g., in-situ offshore groundwater measurements), and new observational and experimental frameworks. These
endeavours should facilitate fully representative parameterizations of FSGD-OFG connectivity in numerical models across the land-sea interface. Moreover, developing hydrologic/oceanographic models of coastal and offshore groundwater and its interactions with other system compartments (sediments, water column, subseafloor environments) will help predicting future changes of groundwater on both regional and global scales.






It is evident that only multidisciplinary research initiatives, at both local, national and international levels, can effectively address the research tasks identified in this perspective paper. Joint projects should link laboratory, field, and modelling approaches to better understand the complex interplay of the various physical, chemical and biological processes operating along the land-ocean interface. Likewise, sustained
observations will help to amend the current uncertainties in temporal variability of groundwater flows. An improved understanding of land-ocean connectivity in this context will contribute to our appreciation of the crucial role of coastal groundwater in societal-relevant issues such as climate change, pollution and the overall environmental status of the coastal oceans. Future research efforts in this topic will directly address the Sustainable Development Goals 6 (*"Clean water and sanitation"*), 12 (*"Responsible*
*consumption and production"*) and 14 (*"Life below water"*) of the United Nations (see https://www.un.org/sustainable development /sustainable-development-goals/).

### *Author contributions*
The initial draft of this perspectives manuscript was prepared by D.L.A.M., A.H., H.W.B, E.E., M.J.,
N.M. and J.S.v.D. All co-authors contributed to the revision and preparation of the final version of the manuscript.

### *Competing interests*
The authors declare that they have no conflict of interest.

### *Acknowledgements*
The manuscript is a contribution to the KiSNet Network funded by the German Research Foundation (DFG; Grant # MA7041/6-1) and the DFG research training group Baltic TRANSCOAST. A.M. has received funding from the European Research Council (ERC) under the European Union's Horizon 2020
research and innovation programme (grant agreement No 677898 (MARCAN)). The conception of this manuscript was fostered by the discussions during a workshop sponsored by the Future Ocean Network of Kiel University (https://www.futureocean.org/en/) in February 2021. D.L.A.M. is supported by the Future Ocean Network (Grant # FON2020-03).

### *Dedication*
*This article is dedicated to Dr. Volker Liebetrau (1965–2022). Volker, our co-author and friend, passed away on February 7th 2022. Volker is remembered for his enthusiasm, commitment, and smile. His passing is a massive loss to the community, but there will be much that will still be carried on in his memory thanks to his hard work and passion.*

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





## Figures

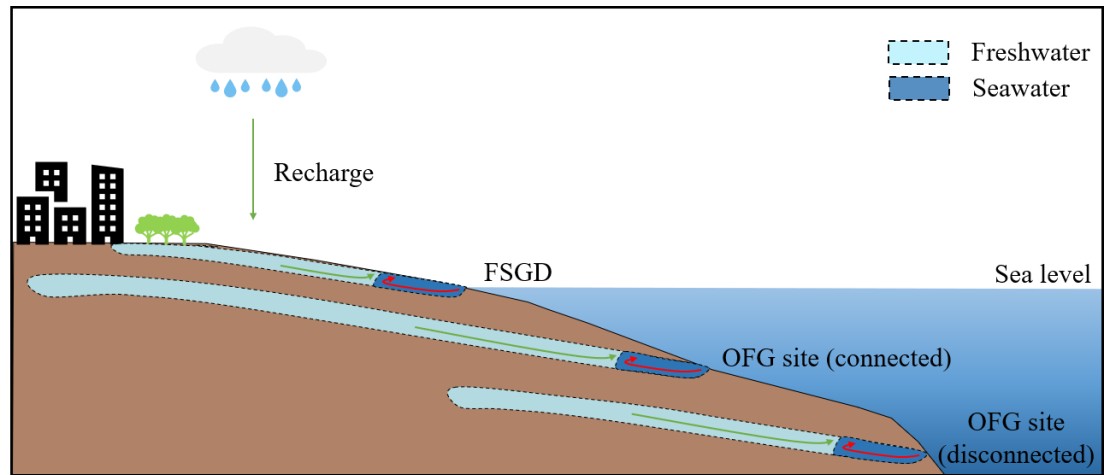

**Figure 1.** Schematic representation of known pathways for the transport of and storage of fresh and freshened groundwater between terrestrial and marine realms. Areas surrounded by dashed lines indicate groundwater reservoirs, whereas arrows represent freshwater (green) and seawater (red) fluxes. Based on Bratton et al. (2010) and Weymer et al. (2020).

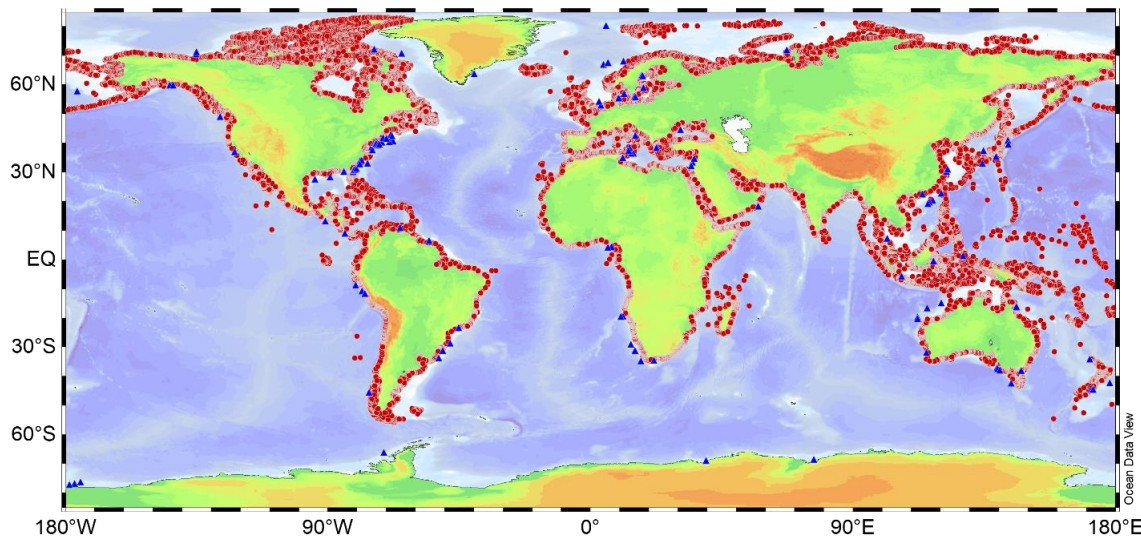

**Figure 2.** Global distribution of reported FSGD (red circles) and OFG (blue triangles) sites. Location data from FSGD and OFG from Luijendijk et al. (2020) and Micallef et al. (2021), respectively.




## Tables


**Table 1.** Key concepts used in this manuscript.

| Term | Definition |
|---|---|
| Meteoric water | Waters derived from precipitation. These waters reach the ocean either through surface flows (e.g. rivers), or as groundwater after infiltrates in soils. |
| Aquifer | Underground water reservoir that can consist of several layers of rock or sediments. |
| Groundwater | Water reservoir located beneath land surfaces. |
| Groundwater recharge | Replenishment of an aquifer containing groundwater from surface sources. |
| Fresh submarine groundwater discharge (FSGD) | Flow of fresh meteoric groundwater from terrestrial coastal aquifers through the seafloor into the ocean. |
| Offshore freshened groundwater (OFG) | Reservoir of fresh and brackish groundwater embedded in sediment pore waters and rocks below the seafloor. |
| Seawater intrusion (SWI) | Flows of marine waters into freshwater aquifers. |
| Non-renewable groundwater | Groundwater whose renewal (through recharge) takes place in times scales > 100 years (see Bierkins and Wada, 2019). |
| Fossil groundwater | Groundwater stored over millennia in isolated reservoirs below the Earth's surface. |
| Subterranean estuary | Coastal aquifer connected to the ocean which bears both saline and meteoric waters. |

**Table 2.** Commonly used methods for investigating groundwater fluxes and reservoirs. [*] Current application realm.

| Approach | Spatial scales | Temporal scales | Captured processes / controlling mechanisms | FSGD/OFG[*] |
|---|---|---|---|---|
| Thermal infrared sensing | cm to km | hours to months | Inflow of low-density plumes. Assessment on sea surface temperature anomalies with respect seasonal means | FSGD |
| Electrical ground conductivity | m to km | hours to years | Temporal variability of fresh-salt interfaces Recirculation fluxes Setting of sub-surface salt balance models | FSGD/OFG |
| Seafloor mapping & Sub-bottom profiling (Acoustics) | cm to km | - | Presence of seafloor depressions (e.g. "Wonky Holes") Pockmarks formation | FSGD/OFG |
| Electromagnetics | m to km | - | Electrical resistivity anomalies within the seafloor and water column that are indicative of active groundwater discharge | FSGD/OFG |
| Direct measurements of seepage rates | cm to m | hours | Quantification of fresh groundwater discharge rates | FSGD |
| In-situ surveys with remotely operated vehicles | m to km | - | Quantification of fresh groundwater discharge rates | FSGD |
| Hydrological modelling | m to km | - | Characterization of groundwater fluxes and chemical transformations Simulation of aquifer properties under hydrological changes | FSGD/OFG |





| Radon isotopes measurements | cm to km | days | Assessment of local sources and recent inputs based on strong gradients between groundwater and ocean | FSGD |
|---|---|---|---|---|
| | | | Tracking of groundwater-derived greenhouse gases | |
| Dissolved organic matter measurements | cm to km | - | Concentration distributions and composition are used to track FSGD properties and dispersal | FSGD |
| Measurements of $\delta^{13}C$ and $\delta^{15}N$ signatures | km | - | Assessment of spatial distribution and C and N flows due to FSGD | FSGD/OFG |
| Nutrient analysis | km | - | Assessment of spatial distribution and estimation of primary production | FSGD/OFG |
| Water isotopes ($\delta D$ and $\delta^{18}O$) | m to km | months to centuries | Identification of recharge processes | FSGD |
| Gas measurements | cm to km | days to months | Assessment of FSGD-driven net community production Quantification of trace gas production and emissions to the atmosphere | FSGD/OFG |
| Phytoplankton analysis | km | - | Assessment of FSGD effects on primary production | FSGD |
| Benthic fauna sampling | m to km | - | Assessment of FSGD effects on benthic biomass & diversity | FSGD |
| Microbial ecology analyses | cm to km | - | Evaluation of abundance and diversity differences within FSGD sites | FSGD |