# Peer review of "Ideas and perspectives: Land-ocean connectivity through groundwater"

_Biogeosciences, 2022_

## Referee Comment (RC1)

[referee-annotated manuscript omitted]

---

## Referee Comment (RC3)

[referee-annotated manuscript omitted]

---

## Author Response (AR1)

**GEOMAR** | Düsternbrooker Weg 20 | 24105 Kiel | Germany

**Dr. D. L. Arévalo-Martínez**
Tel +49 431 600-4207
darevalo@geomar.de

**Biogeosciences Editorial Team**

20. Oktober 2022

Dear Dr. Bond-Lamberty,

hereby I submit a revised version of our manuscript *"Ideas and perspectives: Land-ocean connectivity through groundwater"* to be considered for publication in Biogeosciences.

We are glad to see that our manuscript has been well received and are now invited to upload the newest version. Attached we are sending the marked-up document which includes the changes highlighted in our replies to the community (CC) and reviewers' (RC1, RC2) comments. In addition to those changes, we corrected a few typos which we identified during the review and included two new (2022) relevant references. A detailed list of those changes is provided below (see Annex 1).

We appreciate your attention and look forward to your reply.

Yours faithfully,

Damian L. Arévalo-Martínez

**GEOMAR**
Helmholtz-Zentrum für
Ozeanforschung Kiel

Düsternbrooker Weg 20
24105 Kiel | Germany

Tel +49 431 600-0
Fax +49 431 600-2805
www.geomar.de

Deutsche Bank AG Kiel
BLZ 210 700 24
Kto. 144 8000

SWIFT/BIC DEUTDEDB210
IBAN DE 69210700240144800000

Steuernummer 1929401912
VAT DE281295378

**Stiftung des öffentlichen Rechts**
MinDir Dr. Karl Eugen Huthmacher, *Vorsitzender des Kuratoriums*
Prof. Dr. Peter Herzig, *Direktor* | Michael Wagner, *Verwaltungsdirektor*

[Figure]

[Figure]

**Annex 1 – Additional changes to the manuscript**

These changes were not derived from the community and reviewer's comments, but from the authors' own assessment of the revised version after implementing the replies to those comments.

- l. 4–28 Numbering of affiliations updated to accommodate the new affiliation of the first author
- l. 150 Replaced "from which" by "of which"
- l. 183–184 Ordered citations chronologically
- l. 189–190 Added reference from new study on methane release from pockmarks in association with FSGD (Purkamo et al., 2022)
- l. 221 Replaced "sustained monitoring the" by "sustained monitoring of the"
- l. 258 Changed "coast line" by "coastline"
- l. 319 Changed "model" to "models"
- l. 326 Deleted second occurrence of "offshore"
- l. 334 Deleted "can"
- l. 394 Replaced "identification" by "detection"
- l. 399 Added reference from new study on biogeochemical aspects of FSGD (Ikonen et al., 2022)
- l. 408 Removed capitalization from "Radon" and "Radium"
- Acknowledgements: test modified such that the main grant for the paper is listed first (FON2020-03)
- l. 484–485 Additional acknowledgement to reflect contribution of DFG research group "DynaDeep"
- l. 487 Changed "is" by "was" since DLAM changed institution during the review process of the manuscript
- Reference list

Adjusted position of Liu et al. (2010) and (2017) to keep chronological order

Added: Ikonen, J., Hendriksson, N., Luoma, S., Lahaye, Y., and Virtasalo, J. J.: Behavior of Li, S and Sr isotopes in the subterranean estuary and seafloor pockmarks of the Hanko submarine groundwater discharge site in Finland, northern Baltic Sea, Appl. Geochem., 105471, https://doi.org/10.1016/j.apgeochem.2022. 105471, 2022.

[Figure]

Added: Purkamo,L., Milene, C., von Ahn, E., Jilbert, T., Muniruzzaman, M., Bange, H. W., Jenner, A.-K., Böttcher, M. E., and Virtasalo, J. J.: Impact of submarine groundwater discharge on biogeochemistry and microbial communities in pockmarks, Geochem. Cosmochem. Acta, 334, 14–44, 2022.

**Author's response to community comments (Dr. Clara Ruiz-González)**

*Very relevant and timely article! The authors might consider including a recent review on the microbial dimension of submarine groundwater discharge by Ruiz-González, Rodellas and Garcia-Orellana (2021, FEMS Microbiology Reviews), which evidences the poor knowledge of the (micro)biological aspects related to submarine groundwater discharge, ranging from the microbially-driven chemical transformations of the groudnwater within coastal aquifers to the microbial responses to groundwater inputs once in the coastal ocean. Current challenges and future directions of the field are also highlighted, empashizing, as in the current article, the need for multidisciplinary collaborations. This review article was published slightly earlier than the one of Archana et al (2021), already cited in the text, but targets the entire groundwater-marine continuum by discussing the microbial implications of groundwater discharge in the ocean.*

*Clara Ruiz-González, Valentí Rodellas, Jordi Garcia-Orellana, The microbial dimension of submarine groundwater discharge: current challenges and future directions, FEMS Microbiology Reviews, Volume 45, Issue 5, September 2021, fuab010, https://doi.org/10.1093/femsre/fuab010*

Many thanks for your interest in our manuscript and the suggestion. The (micro)biological aspects of both freshened submarine water discharge and offshore freshened groundwater certainly need further investigation, and their role within future multidisciplinary approaches is of course a topic we would like to highlight with our contribution.

We have included now the suggested publication in the revised version of the manuscript (l. 251–252).

**Author's response to reviewers' comments (bg-2022-148-RC1)**

On behalf of the authors, I thank Reviewer #1 (Dr. Ana Silva) for the positive assessment of our manuscript, as well as the constructive comments and suggestions. On the following we provide a point-by-point response to the issues raised during the review process, and list/discuss the changes done to the revised version:

*Overall this manuscript provides a short summarising view of groundwater as a linking element between land and sea. Its scope is adequate for the publication format and the argumentation is well made in several key points. The topic and scope are very timely and match a research area reaching a cumulative point of becoming useful in management contexts. Albeit of considerable utility to the target readers, the manuscript reads mostly as a summarising text, where the authors' own contributions appear very scattered and hard to find; summarising their original contributions at the end of the sections might be a simple solution to enhance the manuscript's uptake and impact. I made several annotations in the attached pdf to be considered by the authors. Most reflect a need to alter the document to become more reader-friendly and add some contribution ambition, which is somewhat limited in this version. The point made about the common language/proposed framework for existing methodologies would be a particularly desirable adding value.*

Reply by authors:

Many thanks! We are glad to see that our manuscript is seen as useful for researchers from a wide range of expertise. We would like to clarify that with this manuscript we aim to present the readers our view with respect to the impelling need of developing a framework for joint, multidisciplinary studies on FSGD and OFG, rather than offering the framework itself or recommending an optimal combination of methods which should be used. We contend that identifying the current data and process understanding gaps in FSGD/OFG research (as presented in this manuscript) is precisely the author's contribution requested by Reviewer #1.

*l. 74–76  This may be a limited definition as it seems to leave out the groundwater flowing at the surface from coastal aquifers into the intertidal zone. I encourage the authors to add information and arguments positioning this situation in their 2-element classification. for instance see:*
*https://www.sciencedirect.com/science/article/abs/pii/S0141113622001179*

Reply by authors:

Thank you for the suggestion. We disagree in that our definition disregards FSGD into the intertidal zone, since it generally refers to flows of groundwater to the ocean, independent on the level at which this might occur. However, we would like to avoid other readers having the same impression and therefore changed slightly our formulation so that it is clear that we refer to fluxes into the ocean through the coastal zone (which by definition includes the intertidal). We hope this clarifies the confusion. The revised sentence reads as follows: *"The first comprises meteoric groundwater flux from terrestrial aquifers through the seabed (including the intertidal zone) into the coastal ocean, (...)"* (l. 75–76 of revised manuscript).

*l. 103–108 given this self-proposed context I suggest for sections 1+2 have a dedicated subheading focusing on current bottlenecks/limitations/gaps so that the reader can obtain a summary of where we are now in this research, ie, state of the art. This would also help further in justifying the review pertinence.*

*also and importantly, given that the authors claim here existing issues related to subfield-languages heterogeneity, I would expect this review to be greatly imprived by adding the objective of creating a standardised framework integrating the most promising methodologies and proposing a common language. this would enable a practical uptake of this manuscript, enhancing its impact.*

Reply by authors:

Thank you for the suggestions. We would like to clarify that our manuscript purposefully deviates from a review paper, since we think there is already an important number of excellent papers of that type which address the specifics of our current knowledge on e.g. hydrological, geological, geochemical and biological aspects of groundwater fluxes and reservoirs.

We contend that we have included information on bottlenecks/limitations/gaps throughout the different sections of the paper. Considering that these aspects can look different depending on whether we are discussing, for instance, spatial distribution or environmental impact, adding subheadings to sections 1 and 2 (as suggested by Reviewer #1) would imply adding them uniformly through all sections. Including subcategories that apply to all sections is rather unfeasible and we think the result would not be in the best interest of the reader because of an unnecessary increase in the complexity of the document.

Moreover, the main goal of our manuscript is not to provide a unique framework which can be used by all groups conducting research on FSGD and OFG, but rather to convey the need of developing it and provide examples on its added value, should it be used for future multidisciplinary studies. In summary, we hope that our contribution fosters future (international) initiatives for joint investigation of FSGD and OFG.

*l. Section 3 This section requires additional sublevels of subheadings categorization given its extension and scope; as it is, the reading flow becomes very cumbersome*

Reply by authors:

Thank you for the suggestion. However, we respectfully disagree. This section comprises five interrelated paragraphs which follow a logical sequence that would be interrupted by sublevels. Also in this case we do not think that separating the text would help the reader to grasp the arguments presented in this section.

*l. 175–177 This requires further explanation/detail to support the argument pertinence*

Reply by authors:

Thank you for the suggestion. We changed our formulation in order to strengthen our argument. The revised sentence reads as follows: *"In contrast, the role of OFG as a fresh- or brackish water habitat within a purely marine environment remains unknown and might constitute a new frontier in ocean sciences, also in view of its potential exploitation as an unconventional source of water."* (l. 176–178 of revised manuscript).

*l. 360–362 consider adding here also this alternative method based on thermal imagery: https://www.mdpi.com/2077-1312/10/3/414*

Reply by authors:

Thanks for the suggested reference. We have added it, both within the text (l. 365–366 of revised manuscript) and the list of references.

*Section 5 I strongly encourage the authors to provide here either a method-focused summarised figure or table. This section is very long, detailed and for the readers benefit, the authors should finish with their proposed (combination of methods?) solution for different scenarios*

Reply by authors:

Thank you for the suggestion. As explained above, the purpose of our manuscript is not to provide an "ultimate" guideline on how to carry out joint FSGD/OFG research, but rather to identify the current gaps as well as potential ways of cooperation to address those gaps. We are reluctant to provide a "recommended" approach, since to that end a higher level of international coordination would be needed. The realization that activities to achieve that coordination is timely, is precisely what we would like to bring to the community with this manuscript. As for the length, we respectfully disagree since we see no reason why this particular section should be further shortened (it is indeed the shortest of the manuscript).

*l. 466 I strongly encourage the authors to add a "conclusion" section focusing in the impacts of their analyses and discussions in the manuscript, ie, added value of the proposed tasks as a whole, contours/preliminary framework for unifying/merging methodologies and common language approach,*

Reply by authors:

We respectfully disagree. In our opinion, a conclusions section is not appropriate for this type of manuscript because we are not presenting and/or discussing scientific results. Moreover, considering the length of the paper, a summary of the arguments brought about within it would not represent a significant contribution. With our last section (5), we instead hope to convey suggested ways forward which are derived from the knowledge gaps which were discussed in the previous sections.

*l. 1020 For the readers' support I suggest this table to include some example references. I also leave for your consideration this reference: https://www.sciencedirect.com/science/article/abs/pii/S0141113622001179*

Reply by authors:

Thank you for the suggestion. We added the study to the text (l. 184 of revised version) and list of references since it fits well within the context of the manuscript. As for Table 2, example references of the different approaches have been included/discussed in the manuscript. We therefore do not see necessary to include them there. Considering that there are several references that can be used for each method, we opted for avoiding the bias that would imply selecting certain studies and "recommend" them as promising.

Kind regards,

Damian L. Arévalo-Martínez

**Author's response to reviewers' comments (bg-2022-148-RC2)**

On behalf of the authors, I thank Reviewer #2 for the positive assessment of our manuscript, as well as the constructive comments and suggestions. On the following we provide a point-by-point response to the issues raised during the review process, and list/discuss the changes done to the revised version:

*This article provides a summary of the important roles, ideas, and prospects of groundwater as an important link between land and sea. In particular, the authors classify meteoric groundwater into coastal runoff FSGD and offshore freshened OFG, and emphasize the role of each and the connectivity between the two. This manuscript is very timely and necessary as the definitions scattered in several papers were arranged, the latest research from around the world were cited, as well as a perspective or direction for groundwater study is provided. Nevertheless, some suggestions are annotated in the attached pdf to help the reader understand.*

Reply by authors:

Many thanks, we are glad to see that the relevance of our contribution comes across clearly and that the manuscript is seen as useful for researchers from a wide range of expertise.

*l. 147 (1\*10$^6$ km$^3$) Readers may not know what this number means and whether it is important or not. It would be better if a suitable metaphorical example could be provided to give the reader an idea of this quantity.*

Reply by authors:

Thanks for the suggestion. To offer the reader a point of comparison for this number, we added the percentage it would represent in with respect to the estimated total fresh water sources on Earth. The revised sentence reads: *"Recent estimates report OFG to comprise a volume of approximately 1\*106 km3 (Micallef et al., 2021), which is about 10% of the Earth's liquid fresh water (Shlklomanov, 1993)."* (l. 146–148 of revised manuscript). We added the corresponding entry to the list of references.

*l. 218 also in Jeju Island, Korea (Kwon et al., 2017 Scientific reports; Cho et al., 2019 Science of the total environment)*

Reply by authors:

Thanks for the suggestion. We adjusted the sentence to accommodate the additional case studies. The revised sentence reads: "*For example, large outbreaks of the macroalgae Ulva spp. (so-called "green tides"), which occur regularly in eutrophic coasts off China and Korea, are attributed to the nutrient supply by FSGD (Kwon et al., 2017; Liu et al., 2017; Cho et al., 2019; Zhao et al., 2021)."* (l. 218–220 of revised manuscript). We added the corresponding entries to the list of references.

*l. 241–249 For readers support I suggest this part to include some references.*

Reply by authors:

Thanks for the suggestion. We added three references of papers which present comprehensive overviews on the methodologies used for geophysical, hydrological, biogeochemical and microbial approaches to investigate FSGD/OFG (l. 251–252 of revised manuscript). We added an entry to the list of references, since one of the publications was not originally cited in our manuscript:

*Clara Ruiz-González, Valentí Rodellas, Jordi Garcia-Orellana, The microbial dimension of submarine groundwater discharge: current challenges and future directions, FEMS Microbiology Reviews, Volume 45, Issue 5, September 2021, fuab010, https://doi.org/10.1093/femsre/fuab010*

We are thankful to Dr. Ruiz-González, who made us aware of the paper as part of her community comment to our manuscript.

*l. 995 How about displaying it as "FSGD site" rather than FSGD for uniformity?*

Reply by authors:

Thanks for the suggestion. However, in our manuscript we are trying to convey the point that when we refer to FSGD we are talking about fluxes, whereas when we refer to OFG we talk about reservoirs. Adding a reference to "FSGD sites" in this context might lead to confusion. Moreover, the spatial distribution of FSGD occurrences can be seen in Figure 2.

*l. 995 How about additionally displaying the distance scales corresponding to the FSGD site and OFG site?*

Reply by authors:

Thanks for the suggestion. However, we don't see an added value in including the scale distances in this context, since our manuscript is focused on the conceptual aspects of both fields of research, rather than the details of the methodological needs that might arise due to different scales of variability. Thus, adding the scales might unnecessarily complicate the figure for the reader. The different scales of variability are included in some of the review papers which we cited in our manuscript (e.g., Bratton, 2010).

Kind regards,

Damian L. Arévalo-Martínez

---

## Referee Report (RR1)

**EGU Biogeosciences Review**
Manuscript: "*Ideas and perspectives: Land-ocean connectivity through groundwater*"
Recommendation: Accept subject to minor revisions

Review:

This manuscript gives an overview of land-ocean connectivity through groundwater and highlights knowledge gaps in the current knowledge of the connection between meteoric groundwater discharged at the coastline and groundwater discharged offshore. The authors describe methodologies used to characterize flow paths, connectivity, and quantify discharge via groundwater and highlight the difficulties associated with connecting groundwater and its related impacts across the terrestrial-aquatic interface. The authors identify research areas to be addressed in future work to better understand groundwater across spatial and temporal scales. They stress the importance of leveraging interdisciplinary teams in order to accurately work across land and sea boundaries and to fully understand the impacts of groundwater in the face of anthropogenic stresses and a changing climate.

Previous reviewers offered suggestions to clarify the manuscript, add references, and suggest a more compelling conclusion. The authors addressed these suggestions in their response and made changes to clarify the manuscript. They make it clear that this article aims to be a discussion of knowledge gaps rather than a full review of the literature or a detailed description of a framework for future research. I feel this manuscript is suitable for publication in Biogeosciences. I suggest the authors consider only a few minor revisions prior to publication the manuscript, which are outlined below. I feel these minor comments will add to the clarity of the manuscript and make a more citable paper.

Overall comments:
- There are a lot of complex lists and parentheses throughout the manuscript that can make it hard for the reader. It may be worth varying sentence structure a bit and breaking up some long sentences.
- It was suggested by a previous reviewer to have a 'conclusions' section. I found the authors rebuttal to adding this section adequate, but I would urge the authors to consider adding a few sentences that add value to their recommended future research areas. Although it is clear the authors do not hope to provide a framework for how to conduct interdisciplinary groundwater research, simply identifying the need for interdisciplinary work does not feel like a very novel conclusion or suggestion for the field. Perhaps the instead of ending with the need for interdisciplinary teams and then how this work will address the sustainable development goals, this last section could be rearranged to conclude with reiterating the impacts of groundwater globally and why these research areas and interdisciplinary action to address them are so vital to understanding the global ocean. The sustainable development goals could provide structure for these suggested few sentences to re-establish the importance and take-home point of the paper.

Specific Comments:
- Line 58: The sentence "Coastal margins play a disproportionally important role for productive marine ecosystems compared to the open ocean due to their greater biological productivity, sediment-water…" reads awkwardly. I suggest removing the first word 'productive' and simply say "important role for marine ecosystems"

- Line 75: The sentence, "The first comprises meteoric groundwater flux from terrestrial aquifers through the seabed (including the intertidal zone) into the coastal ocean,…" has awkward wording. Perhaps "The first is comprised of the meteoric groundwater flux from terrestrial aquifers.."
- Line 146: "OFG resides beneath the seafloor along continental shelves and, in contrast to FSGD, is commonly assumed to have minimal groundwater flow velocities (e.g. Micallef et al. 2020)." Perhaps "have low groundwater flow velocities"
- Line 417: "FSGD and the associated fluxes of biogeochemical tracers might affect the physical structure, chemical composition and reactivity and the (micro)biology of the coastal ocean ecosystems." I would add a comma after reactivity and I would remove "the" before coastal ocean ecosystems
- Figure 1: The Arrows in figure 1 are very small and hard to see – I didn't realize they were arrows at first. Perhaps they could be made a different color that stands out or made larger.

---

## Author Response (AR2)

GEOMAR | Düsternbrooker Weg 20 | 24105 Kiel | Germany

**Dr. D. L. Arévalo-Martínez**
darevalo@geomar.de

**Biogeosciences Editorial Team**

January 27th 2023

Dear Dr. Bond-Lamberty,

hereby I submit a revised version of our manuscript *"Ideas and perspectives: Land-ocean connectivity through groundwater"* to be considered for publication in Biogeosciences.

We are glad to hear that you and the two new referees have a positive opinion of our manuscript, and that our revised version has addressed well the comments provided by the referees during the first round.

Attached we are sending the marked-up document which includes the changes highlighted in our replies to both referees' comments. In addition to those changes, we made a few minor modifications which we saw necessary after reading the latest version of our paper. A detailed list of those changes is provided below (see Annex 1). We hope with these improvements the manuscript can be approved for publication.

We appreciate your attention and look forward to your reply.

Yours faithfully,

Damian L. Arévalo-Martínez

**GEOMAR**
Helmholtz-Zentrum für
Ozeanforschung Kiel

Düsternbrooker Weg 20
24105 Kiel | Germany

Tel  +49 431 600-0
Fax +49 431 600-2805
www.geomar.de

Deutsche Bank AG Kiel
BLZ 210 700 24
Kto. 144 8000

SWIFT/BIC DEUTDEDB210
IBAN DE 69210700240144800000

Steuernummer 1929401912
VAT DE281295378

**Stiftung des öffentlichen Rechts**
MinDir Dr. Karl Eugen Huthmacher, *Vorsitzender des Kuratoriums*
Prof. Dr. Peter Herzig, *Direktor* | Michael Wagner, *Verwaltungsdirektor*

[Figure]

[Figure]

**Annex 1 – Comments for Editor (Dr. Ben Bond-Lamberty)**

These changes were not derived from the referee's comments, but from the authors' own assessment of the revised version after implementing the replies to those comments.

- We updated the numbering of affiliations of all authors to accommodate the new affiliation of the first author (l. 4–28).

- We removed "climate change" from the first sentence to avoid confusion since it would give the impression that this is related to land only (see l. 230).

- We removed the space between "ground" and "water" (l. 307).

- We exchanged "model" by "models" (l. 321).

- We removed the word "can" since it was not relevant in this context (l. 336)

- We removed extra parenthesis in the citation of Worzewski et al. (l. 347)

- We added the word "dynamics" for more clarity (l. 461)

- We slightly changed one sentence in the acknowledgements: from "This manuscript is a contribution to" to "This manuscript contains contributions to", added a missing acknowledgement (SMART Project), and changed the position of one of the acknowledged projects in the text. (see l. 487–492).

- Previous versions of the manuscript included a work in press (book chapter by Böttcher et al.). In the revised version we added the missing information (see l. 549–551).

- We added a missing year on the citation of the work by Church et al. (see l. 594–598).

- Following the editor's and reviewer's suggestions, we modified Figure 1 for more clarity.

**Author's response to comments by anonymous referee #3**

On behalf of the authors, I thank anonymous referee #3 for the positive assessment of our manuscript and the constructive comments and suggestions. On the following we provide a point-by-point response to the issues raised during the review, and list/discuss the changes done to the revised version:

*This manuscript provides a brief overview of coastal groundwater dynamics, which is an important, timely, and understudied topic. The main message of the manuscript seems to be that more synergy is needed between research studying fresh submarine groundwater discharge and offshore freshened groundwater discharge and techniques applied in each domain should be merged in future interdisciplinary efforts. This is a notable cause for the research community to take on.*

Reply by authors:

Many thanks. We are glad to hear that our manuscript conveys the need of taking on international multidisciplinary efforts for investigating coastal groundwater dynamics. We hope other readers will also share this opinion and that our manuscript serves as starting point for fruitful discussions in the community.

*Overall, the manuscript gives a nice high-level overview of the topic and is generally approachable to a general audience, though at first the frequent use of acronyms is a bit jarring. Having table 1 shown in the first several pages in the final version will help this, as I found myself needing to look up the difference between FSGD and OFG several times while reading the introduction and section 2.*

Reply by authors:

We fully agree. In the current version we refer to Table 1 on the second paragraph of the introduction and are therefore confident that the editorial team will identify the right place for it in the first pages of the manuscript.

*This review details mostly conceptual information and highlights key methods used in this domain of research. The study could be substantially more impactful if some form of quantitative analysis or summarization was performed such as a table showing ranges in discharge yields by geological setting, etc. However, I understand that such analyses would require substantial effort and is probably outside the scope of this short perspective paper. Considering the intended purpose of this perspective (i.e., high level overview of the field of research and call to action) I only have several specific comments below. The authors have done a nice job responding to previous reviews and I'm supportive of its publication.*

Reply by authors:

Many thanks for your constructive comments. We indeed opted for avoiding a quantitative analysis in the framework of this manuscript, because there are excellent review papers which address our current knowledge on hydrological, geological, geochemical and biological aspects of groundwater fluxes and reservoirs. Some of the examples cited in our manuscript are:

Micallef, A., et al.: Offshore freshened groundwater in continental margins, Rev. Geophys., 58, e2020RG000706, https://doi.org/10.1029/2020RG000706, 2021.

Ruiz-González, C., et al.: The microbial dimension of submarine groundwater discharge: current challenges and future directions, FEMS Microbiol. Rev., 45(5), fuab010, https://doi.org/10.1093/femsre/fuab010, 2021.

Santos, I. R., et al.: Submarine groundwater discharge impacts on coastal nutrient biogeochemistry, Nat. Rev. Earth. Environ, 2, 307 – 323, https://doi.org/10.1038/s43017-021-00152-0, 2021.

Taniguchi, M., et al.: Submarine Groundwater Discharge: Updates on Its Measurement Techniques, Geophysical Drivers, Magnitudes, and Effects, Front. Environ. Sci., 7, 141, 2019.

*Figure 1: I like the simple figure, but there are some confusing details if you look closely beyond the fact that FSGD happens nearshore and OFG happens offshore. For example, should saltwater intrusion also be shown in this figure, or is that implied by the dark blue parts of FSGD and OFG and red arrows? On that note, with the current design, it actually doesn't look like the light blue freshwater ever reaches the sea. I'm not sure what the solution is, perhaps more of a color gradient? Salt water intrusion can also occur in the shallower terrestrial subsurface while fresh groundwater is discharged deeper/further offshore. Also, perhaps a dashed line could also represent rising sea levels? There are just suggestions should you have ideas how to improve on these nuances. With some improvements, the figure would be very useful for conceptual and educational purposes. As noted in my general comments, considering this perspective paper provides a high level overview, a very nice conceptual figure that gets widely used and shared would elevate the paper's impact and influence.*

Reply by authors:

This is a very good point and we are glad it was brought to our attention. The revised Figure 1 includes a colour gradient that illustrates the progressive change in the dominance between fresh and saltwater along the land-ocean transition and different topographic levels. Moreover, we increased the thickness of the aquifers to improve visibility, and made sure the interfaces between fresh and saltwater components are kept as realistic as possible for such a schematic representation. These modifications supplement the green and red arrows which are now in more places and are thicker for better visibility.

*Line 40 and throughout: What is the reason for using parentheses for (bio)geochemists here and a few other places?*

Reply by authors:

We introduced this notation in order to highlight that the investigation of coastal groundwater dynamics will require expertise not only from biogeochemists, but also from inorganic geochemists. We preferred this short version instead of mentioning both groups separately every time.

*Line 201: The influence of pumping practices and local reservoir properties on saltwater intrusion are astutely pointed out in this paragraph. Not much is said about sea level rise impacts on SWI aside from broad discussion of global rates of in the introduction. It might be worth adding some content either in the introduction or this section pointing out that rates of relative sea level rise vary substantially due to local factors such as subsidence and sedimentation. Local relative SLR is what influences saltwater intrusion, not global average rates.*

Reply by authors:

Many thanks for the suggestion; this is an important point to highlight in the context of our manuscript. We added a sentence to address this issue in the revised version of the manuscript (see l. 66–69). The work cited in this context was added to the list of references (see l. 831–833).

*Line 430: This may be implicit in point #3 as well as others, but I wonder if a task related to scaling, specifically would be useful? For example, once these various tasks are achieved will we be able to understand submarine discharge impacts on global scale processes, or is an additional task needed such as representing these fluxes in global Earth system models?*

Reply by authors:

Thanks for the suggestion. We included an additional task that addresses this point (l. 451–452):

"(7) implement the knowledge gained through models and observations to improve the representation of FSGD and OFG in Earth System models."

To keep the logical sequence of arguments for the reader, we shifted the position of tasks 4–6 (see l. 442–452).

Kind regards,

Damian L. Arévalo-Martínez

**Author's response to comments by anonymous referee #4**

On behalf of the authors, I thank anonymous referee #4 for the positive assessment of our manuscript and the constructive comments and suggestions. On the following we provide a point-by-point response to the issues raised during the review, and list/discuss the changes done to the revised version:

*This manuscript gives an overview of land-ocean connectivity through groundwater and highlights knowledge gaps in the current knowledge of the connection between meteoric groundwater discharged at the coastline and groundwater discharged offshore. The authors describe methodologies used to characterize flow paths, connectivity, and quantify discharge via groundwater and highlight the difficulties associated with connecting groundwater and its related impacts across the terrestrial-aquatic interface. The authors identify research areas to be addressed in future work to better understand groundwater across spatial and temporal scales.*
*They stress the importance of leveraging interdisciplinary teams in order to accurately work across land and sea boundaries and to fully understand the impacts of groundwater in the face of anthropogenic stresses and a changing climate.*
*Previous reviewers offered suggestions to clarify the manuscript, add references, and suggest a more compelling conclusion. The authors addressed these suggestions in their response and made changes to clarify the manuscript. They make it clear that this article aims to be a discussion of knowledge gaps rather than a full review of the literature or a detailed description of a framework for future research. I feel this manuscript is suitable for publication in Biogeosciences. I suggest the authors consider only a few minor revisions prior to publication the manuscript, which are outlined below. I feel these minor comments will add to the clarity of the manuscript and make a more citable paper.*

Reply by authors:

Many thanks for the positive assessment. We are glad to hear that our revision addressed the issues raised during the first round of review, and that our suggested approaches for advancing in joint FSGD-OFG research come across clearly.

*There are a lot of complex lists and parentheses throughout the manuscript that can make it hard for the reader. It may be worth varying sentence structure a bit and breaking up some long sentences.*

Reply by authors:

Thanks for the suggestion. We have revised the text thoroughly to check for sentences that needed improvement in this regard.

*It was suggested by a previous reviewer to have a 'conclusions' section. I found the authors rebuttal to adding this section adequate, but I would urge the authors to consider adding a few sentences that add value to their recommended future research areas. Although it is clear the authors do not hope to provide a framework for how to conduct interdisciplinary groundwater research, simply identifying the need for interdisciplinary work does not feel like a very novel conclusion or suggestion for the field. Perhaps the instead of ending with the need for interdisciplinary teams and then how this work will address the sustainable development goals, this last section could be rearranged to conclude with reiterating the impacts of groundwater globally and why these research areas and interdisciplinary action to address them are so vital to understanding the global ocean. The sustainable development goals could provide structure for these suggested few sentences to re-establish the importance and take-home point of the paper.*

Reply by authors:

Thanks for the constructive suggestion. Nevertheless, we contend that the current structure of section 5 is adequate, since for us it is important to first reiterate the importance of FSGD/OFG within the context of basic and applied (societal relevant) research. Only after that is clear, we can take the reader through the tasks which we think are critical for improving our understanding of groundwater dynamics in coastal systems. We argue that the main contribution of the manuscript is not merely identifying the need for multidisciplinary research, but rather joint efforts to address what in our opinion are urgent research tasks. A detailed discussion on how each of the specific tasks links to Sustainable Development Goals is certainly a logical step; however it falls beyond the scope of our manuscript. We hope that readers will be motivated to elaborate on such links as part of future projects.

*Line 58: The sentence "Coastal margins play a disproportionally important role for productive marine ecosystems compared to the open ocean due to their greater biological productivity, sediment-water…" reads awkwardly. I suggest removing the first word 'productive' and simply say "important role for marine ecosystems"*

Reply by authors:

Changed as suggested (see l. 58–60).

*Line 75: The sentence, "The first comprises meteoric groundwater flux from terrestrial aquifers through the seabed (including the intertidal zone) into the coastal ocean,…" has awkward wording. Perhaps "The first is comprised of the meteoric groundwater flux from terrestrial aquifers.."*

Reply by authors:

Thanks for the suggestion. However, we prefer to keep the current sentence because it has an active rather than a passive voice that fits better to the style in which the manuscript is written.

*Line 146: "OFG resides beneath the seafloor along continental shelves and, in contrast to FSGD, is commonly assumed to have minimal groundwater flow velocities (e.g. Micallef et al. 2020)." Perhaps "have low groundwater flow velocities"*

Reply by authors:

Changed as suggested (see l. 142).

*Line 417: "FSGD and the associated fluxes of biogeochemical tracers might affect the physical structure, chemical composition and reactivity and the (micro)biology of the coastal ocean ecosystems." I would add a comma after reactivity and I would remove "the" before coastal ocean ecosystems*

Reply by authors:

Thanks for the suggestion. Changed accordingly (see l. 418–420).

*Figure 1: The Arrows in figure 1 are very small and hard to see – I didn't realize they were arrows at first. Perhaps they could be made a different color that stands out or made larger.*

Reply by authors:

Thanks for raising this issue. The updated version of Figure 1 has thicker and more abundant arrows which are easier to see. Additionally, we increased the thickness of the aquifers and applied a color gradient so that it is clearer what are the groundwater reservoirs and fluxes we aim to illustrate. We hope that this change (in addition to the changes suggested by anonymous referee #3) make now clearer the features we aim to show with the schematic.

Kind regards,

Damian L. Arévalo-Martínez